# Pre-Operative Frailty Status Is Associated with Cardiac Rehabilitation Completion: A Retrospective Cohort Study

**DOI:** 10.3390/jcm7120560

**Published:** 2018-12-17

**Authors:** Dustin E. Kimber, D. Scott Kehler, James Lytwyn, Kevin F. Boreskie, Patrick Jung, Bryce Alexander, Brett M. Hiebert, Chris Dubiel, Naomi C. Hamm, Andrew N. Stammers, Mekayla Clarke, Carly Fraser, Brittany Pedreira, Navdeep Tangri, Jacqueline L. Hay, Rakesh C. Arora, Todd A. Duhamel

**Affiliations:** 1Institute of Cardiovascular Sciences, St. Boniface Hospital Albrechtsen Research Centre, Winnipeg, MB R2H 2A6, Canada; dustin_kimber@hotmail.com (D.E.K.); scott.kehler@dal.ca (D.S.K.); umlytwyj@myumanitoba.ca (J.L.); kboreskie@sbrc.ca (K.F.B.); patrick.jung@mail.utoronto.ca (P.J.); brycealexander204@gmail.com (B.A.); bhiebert3@sbgh.mb.ca (B.M.H.); dubielc@myumanitoba.ca (C.D.); astammers@shaw.ca (A.N.S.); mekayla_clarke@hotmail.com (M.C.); frasercarly89@gmail.com (C.F.); bpedreira@sbgh.mb.ca (B.P.); ntangri@sbgh.mb.ca (N.T.); umhayj@myumanitoba.ca (J.L.H.); rarora@sbgh.mb.ca (R.C.A.); 2Health, Leisure and Human Performance Research Institute, Faculty of Kinesiology and Recreation Management, University of Manitoba, Winnipeg, MB R3T 2N2, Canada; 3Max Rady College of Medicine, University of Manitoba, Winnipeg, MB R3E 3P5, Canada; 4Department of Community Health Sciences, University of Manitoba, Winnipeg, MB R3E 0W3, Canada; lettn@myumanitoba.ca; 5Department of Surgery, Rady Faculty of Health Sciences, University of Manitoba, Winnipeg, MB R3A 1R9, Canada

**Keywords:** cardiac surgery, cardiac rehabilitation, frailty

## Abstract

While previous investigations have demonstrated the benefit of cardiac rehabilitation (CR) on outcomes after cardiac surgery, the association between pre-operative frailty and post-operative CR completion is unclear. The purpose of this retrospective cohort study was to determine if pre-operative frailty scores impacted CR completion post-operatively and if CR completion influenced frailty scores in 114 cardiac surgery patients. Frailty was assessed with the use of the Clinical Frailty Scale (CFS), the Modified Fried Criteria (MFC), the Short Physical Performance Battery (SPPB), and the Functional Frailty Index (FFI). A Mann-Whitney test was used to compare frailty scores between CR completers and non-completers and changes in frailty scores from baseline to 1-year post-operation. CR non-completers were more frail than CR completers at pre-operative baseline based on the CFS (*p* = 0.01), MFC (*p* < 0.001), SPPB (*p* = 0.007), and the FFI (*p* < 0.001). A change in frailty scores from baseline to 1-year post-operation was not detected in either group using any of the four frailty assessments. However, greater improvements from baseline to 1-year post-operation in two MFC domains (cognitive impairment and low physical activity) and the physical domain of the FFI were found in CR completers as compared to CR non-completers. These data suggest that pre-operative frailty assessments have the potential to identify participants who are less likely to attend and complete CR. The data also suggest that frailty assessment tools need further refinement, as physical domains of frailty function appear to be more sensitive to change following CR than other domains of frailty.

## 1. Introduction

By the year 2051, it is estimated that 25% of the Canadian population will be ≥65 years of age [1]. With an aging population, the incidence of cardiovascular disease is expected to rise, leading to an increased number of cardiac surgical procedures. Improvements in cardiac surgical techniques over the past two decades have decreased operative mortality rates despite an older and increasingly frail cardiac surgery population [2]. Frailty can be defined as the dysregulation of multiple physiological systems, which render the individual vulnerable to health stressors due to a decreased physiological reserve [3]. Currently, more than half of cardiac surgery patients are frail, which places these individuals at an increased risk for post-operative complications [4], including post-operative mortality, morbidity, functional decline, and major adverse cardiac and cerebrovascular events [5]. 

The gold standard for recovery following cardiac surgery is participation in cardiac rehabilitation (CR). CR is a multidisciplinary approach that involves behavior change, risk factor control, exercise, psychological support, and diet education [6]. Those who complete a CR program following their cardiac surgery have improved lipid profiles and reduced risk of all-cause and cardiac mortality when compared to those who do not [7,8]. Unfortunately, CR is underutilized among the cardiac surgery cohort with only 35% starting the program post-surgically [9]. Although the issue of CR underutilization is likely multifactorial in nature, a recent call to action by the European Association of Preventative Cardiology [10] emphasized the paucity of literature analyzing the association between frailty and CR completion. There is a need to understand if patients with frailty attend CR. Moreover, there is a need to determine if frailty is modifiable by CR. Such information may inform the refinement of services to better support adults with frailty as they seek to recover from their cardiac surgery. If attendance can be predicted by frailty, then targeted interventions can be utilized to promote attendance amongst the frail population. Improved integration of frailty into the Canadian healthcare system is needed [11], and research of this type aligns with the top ten priorities for research recently identified by the Canadian Frailty Network [12], as there is an identified need to determine if physical activity can slow the progression, or reverse, frailty. The purpose of this study was to determine the impact of pre-operative frailty on CR completion rates. We hypothesized that pre-operative frailty negatively impacts CR completion rates. 

## 2. Materials and Methods

### 2.1. Trial Design

This study used a retrospective cohort study design among cardiac surgery patients to determine the ability of four commonly used frailty measures to predict CR completion post-operatively.

### 2.2. Ethics and Study Population

The study was approved by the University of Manitoba Health Research Ethics Board (HREB), the Research Review Committee at the St. Boniface General Hospital and the CR Research Review Committee for the regional CR program in Manitoba. Participants were included in this study if they met the following eligibility criteria: (1) ≥18 years of age, (2) undergoing either elective or urgent coronary artery bypass graft (CABG) and/or valve procedures, (3) admitted to the St. Boniface Hospital Intensive Care Cardiac Surgery unit for post-operative care, and (4) able to speak and understand English. Patients were excluded if they had dementia, hearing disabilities, or could not verbally communicate in English. Cardiac surgery patients within this retrospective cohort study were initially enrolled as part of a previous study protocol [4]. This manuscript was developed in congruence with the STROBE guidelines [13].

### 2.3. Cardiac Rehabilitation Program

CR in Manitoba is based on guidelines outlined by the Canadian Association of Cardiovascular Prevention and Rehabilitation in the Canadian Guidelines for Cardiac Rehabilitation and Cardiovascular Disease Prevention: Translating Knowledge into Action 3rd Edition [14]. Briefly, the 16-week center-based program involves a multi-disciplinary team providing exercise and education training to the CR participants. The program is supplemented with optional additional education and voluntary exercise sessions.

### 2.4. Measurements and Outcomes

#### 2.4.1. Frailty Assessment 

Frailty status was measured pre-operatively and 1-year post-operatively in the original prospective cohort study [4]. Since the definition of frailty remains controversial [15], four frailty assessment tools were included in this study to contribute to our overall understanding of frailty’s role in CR completion. The four frailty tools employed in this study included (Table 1): (1) the Clinical Frailty Scale (CFS) [16], (2) the Modified Fried Criteria (MFC) [17,18], (3) the Short Physical Performance Battery (SPPB) [19], and (4) the Functional Frailty Index (FFI; Table A1). Originally, the Fried phenotype model was composed of five criteria focusing on mainly physical variables. The literature has since suggested the inclusion of cognitive and depression criteria to generate a more comprehensive model of frailty, the MFC [18]. The FFI was created by our research group for the context of this study and is based on the frailty index put forth by Mitnitski et al. [20]. The FFI places an emphasis on 25 deficits that may be modifiable with physical activity, scored along a continuum of 0 (no deficit present) to 1 (full expression of the deficit). Focusing on variables that may be modifiable with physical activity, a large component of CR, will optimize the ability of the tool to capture exercise-induced changes in frailty [21].

Furthermore, we analyzed the individual domains of the MFC (shrinking, weakness, exhaustion, slowness, low physical activity, depression and cognitive impairment), SPPB (5-meter gait speed, balance and repeated chair stand) and FFI (physical, functional, nutrition and exhaustion, quality of life and mood, and cognition) to determine how CR completion influenced changes within these frailty domains, as well as the frailty tool that may be most sensitive to change over time. 

#### 2.4.2. Primary Outcome

The primary outcome assessed in this study was CR completion. CR completion, in alignment with the Canadian Cardiovascular Society’s definition, was defined as those individuals who attended a baseline stress test (modified Bruce protocol), attended >1 CR class throughout the program duration and had a formal re-assessment at the program conclusion [22]. CR non-completion was defined as those individuals who did not attend either the baseline stress test, the formal re-assessment at the CR program conclusion or attended ≤1 CR class throughout the program duration [23]. Electronic swipe card access to the facility hosting the center-based CR program was used to assess program attendance. These records were extracted and analyzed for the purposes of the primary and secondary outcome variables. 

#### 2.4.3. Secondary Outcomes

The secondary outcomes: (1) the comparison of change in frailty score from baseline to 1-year post-operatively based on CR completion; (2) the modification of any frailty domain within the MFC, SPPB or FFI based on CR completion; and (3) impact of CR attendance, measured by electronic swipe card attendance records, on frailty score throughout the CR program duration. We defined a CR attender as all individuals who attended ≥1 CR class. To clarify, all CR completers and non-completers who attended ≥1 CR class were considered CR attenders. 

### 2.5. Statistical Analysis

Statistical analyses were performed using TIBCO^®^ Statistica™ (version 13, Palo Alto, CA, USA). The baseline characteristics of CR completers and non-completers, in addition to CR attenders and non-attenders, were compared using the Mann-Whitney test for continuous variables and Chi-Square test for categorical variables. The change in frailty score over time (1-year post-operative frailty score—baseline frailty score) was defined as ∆frailty. Differences between CR completers and CR non-completers in ∆frailty and the domains within were analyzed using a Mann-Whitney test as were frailty scores for CR completers and non-completer 1-year post-operatively. The Spearman Rank Correlation Coefficient was calculated to determine the association between CR attendance and frailty score at baseline and 1-year post-operatively. We defined the Spearman Rank Correlation Coefficient strength based on the following criteria: 0.00 to 0.25 as little to no correlation, 0.25 to 0.50 as a fair correlation, 0.50 to 0.75 as a moderate to good correlation and >0.75 as a good to excellent correlation [24]. A *p*-value of ≤0.05 was determined to be statistically significant. 

## 3. Results

### 3.1. Baseline Characteristics

A total of 235 participants were recruited for the original prospective cohort study between July 2012 and June 2013 (Figure 1) [4]. A median of 51 (38–66) days elapsed between hospital discharge and first attendance at CR. Participants had a median 261 (226–321) days between there last CR attendance and their 1-year post-operative follow up assessment. Of the initial 235 participants, 121 participants were excluded due to not attending the 1-year post-operative follow up assessment or due to completing their follow up assessment over the phone, preventing the collection of objective frailty measures. The excluded patient population had higher prevalence of diabetes than the included group (*p* = 0.01). There was no difference in the type of surgical procedures received between included and excluded participants. Differences in baseline demographics between CR completers and CR non-completers identified higher prevalence of diabetes (*p* = 0.006) and chronic obstructive pulmonary disorder (*p* = 0.04) in CR non-completers (Table 2). Moreover, CR non-completers were significantly more likely to live alone (*p* = 0.02) and have a longer length of hospital stay (*p* = 0.002) when compared to CR completers. 

### 3.2. Pre-Operative Frailty and Post-Operative CR Completion

CR non-completers were significantly more frail than CR completers at baseline based on the CFS (*p* = 0.01), MFC (*p* = 0.0005), SPPB (*p* = 0.007) and FFI (*p* < 0.001; Figure 2). These results are supported by a second approach, where multivariable regression models identified increased odds of not completing CR with every frailty score point increase for the MFC, CFS, and FFI, even after controlling for both age and EuroSCORE II (Table A2). 

### 3.3. Changes in Frailty

Among the four frailty measures analyzed (CFS, *p* = 0.90; MFC, *p* = 0.70; SPPB, *p* = 0.06; FFI, *p* = 0.07), the ∆frailty was not significantly different between CR non-completers and completers (Figure 3). 

### 3.4. Changes in Frailty Domains

The change in the individual MFC frailty domains of slowness, weakness, self-reported weight loss over the past year, exhaustion and depression were not different among CR completers and non-completers (Table 3). However, the change in cognitive impairment (*p* = 0.005) and the change in physical activity (*p* = 0.04) showed greater improvements in CR completers when compared to non-completers. There were no statistically significant changes observed in any domain within the SPPB among CR completers and non-completers. A greater change over time was detected (*p* = 0.009) in CR completers for the physical domain of the FFI, which included five variables: balance score, 5-meter gait speed, chair stand test, timed up-and-go test, and level of physical activity. 

### 3.5. CR Attendance and Frailty

CR attendance was negatively correlated (*r_s_* = −0.29) with baseline frailty as assessed by the CFS (*p* = 0.02; Table 4). No other correlations between CR attendance and frailty measures among CR attenders were significant (Table 4).

## 4. Discussion

The primary purpose of this retrospective cohort study was to determine the impact of pre-operative frailty on CR completion rates. The association between pre-operative frailty and post-operative CR completion was previously not known. Our novel data indicate that pre-operative frailty significantly reduces the likelihood of CR completion post-cardiac surgery. These results were supported by multivariable regression models, where a 1-point increase in the MFC, CFS, or FFI was associated with odds ratios of 0.68 (0.52–0.88 95% CI, *p* = 0.03), 0.65 (0.44–0.96 95% CI, *p* = 0.03), and 0.41 (0.26–0.67 95% CI, *p* < 0.01), respectively, for completing CR after controlling for age and EuroSCORE II. The lower likelihood of frail cardiac surgery patients attending CR is problematic because cardiac surgery patients who are frail pre-operatively are at an increased risk of post-operative mortality, morbidity, functional decline, as well as major adverse cardiac and cerebrovascular events post-cardiac surgery when compared to robust patients [5]. This is an important finding that requires further investigation through larger multisite observational trials to confirm the association between pre-operative frailty and post-operative CR attendance and completion. Even so, the data suggest that there is a need to enhance our understanding of frail cardiac surgery patients to identify factors to motivate attendance and completion. A patient-oriented approach could be implemented for this purpose. Such an approach should utilize the health research roadmap developed by the Canadian Institutes of Health Research [25] and should align with the top ten priorities for research recently identified by the Canadian Frailty Network [12].

CR non-completers, were significantly more likely to have a longer length of hospital stay, live alone, and have more comorbid conditions at baseline when compared to CR completers. It is well documented in the literature that frailty leads to increased length of hospital stay among cardiac surgery patients [26,27,28]. CR non-completers were significantly more likely to live alone when compared to CR completers, which would add further complexity to the subsequent hospital discharge. Living alone is a predictor of 30-day hospital readmission rates, with those living alone experiencing a 3-fold increased risk of readmission following a CABG procedure [29]. Our data suggest a link between frailty and living alone which is also consistent with the literature [30,31]. 

Completion of CR programming is associated with improved lipid profile, reduced hospital readmission and a reduced risk of all-cause and cardiac mortality [8,23,32]. Thus, frail individuals may have the most to gain from completing CR programming post-surgically [33]. It is in this context that the assessment of frailty prior to cardiac surgery may prove to be most beneficial, as the implementation of frailty screening prior to cardiac surgery will provide an opportunity to utilize recruitment strategies tailored for individuals with frailty. In fact, our data indicate there is a need to develop strategies to help mitigate the unique barriers that frail older adults experience with CR attendance and completion. Future research should develop approaches that better support individuals with frailty to attend and complete CR post-cardiac surgery. Alternatively, CR programs need to recognize the opportunity to refine their programming to better meet the needs of patients with frailty and facilitate greater completion rates, and improved health as a result, in this vulnerable population. This initiative would align with the Canadian Frailty Network’s identified need for finding means of improving the health and quality of life for people living with frailty [12]. 

### 4.1. CR and ∆Frailty

The secondary purpose of this study was to determine if CR completion influenced frailty status. The data demonstrate that there was no change in frailty score over time for CR completers or non-completers, with any of the four frailty tools used. Non-completers would not have received the potential benefit of participating in CR, but the lack of frailty modification among CR completers may suggest: (1) CR programming may be ineffective for modifying frailty to an extent that persists in 1-year post-cardiac surgery; or (2) current tools used to measure frailty are not sensitive enough to changes in frailty status over time. Future investigations should consider both possibilities. It is also possible that the better baseline frailty scores of the CR completers resulted in no change over the course of the program due to a ceiling effect in the scores. 

It is important that frailty assessment tools are sensitive to clinically significant changes as there is an emerging recognition that exercise interventions have at least some efficacy for the treatment of frailty [14,34]. Given the dynamic and multifactorial nature of frailty, an appropriate frailty assessment tool should measure multiple frailty components and be rooted in robust evidence [35]. To better identify changes in frailty due to cardiac rehabilitation, researchers should consider the domains of frailty that the tool assesses. For example, multidimensional frailty scores that include physical functioning assessments have the strongest association and largest additional predictive performance for mortality outcomes [36]. A fundamental issue with current measures of multicomponent frailty assessment is that they are often cumbersome and time-consuming to implement in a clinical setting. Financial and time constraints in the healthcare setting mean that frailty assessments should be both valid and feasible for health care practitioners to deliver. The volume of frailty measurement tools in the literature and the absence of a gold standard measure is also problematic [37,38]. Therefore, stakeholders are left to select which frailty tool strikes the best balance between assessing risk for adverse outcomes and feasibility for their given population and context. 

### 4.2. Frailty Domains

In order to further refine existing frailty assessment techniques, we conducted a component analysis of frailty domains for each of the four frailty tools utilized in this study. In our analysis, the MFC domain of low physical activity and cognition showed greater improvement in CR completers compared to non-completers. Declines in cognition measured using the MCOA were significantly lower in the CR completers (*p* = 0.005), which suggests that CR completion may slow cognitive decline in older adults with CVD and frailty. However, that observation remains to be confirmed in larger samples. The higher physical activity levels among CR completers compared to CR non-completers may help explain this finding given that physical activity has been shown in the literature to improve certain aspects of cognitive function and minimize overall decline [39,40,41]. The positive impact of physical activity on cognition is crucial considering that cognitive decline is common among those with CVD and is accelerated among those that are frail [33,42,43].

The data also identified that none of the three SPPB domains of frailty (e.g., 5-meter gait speed, balance, and repeated chair stand) demonstrated significant changes 1-year post-operatively in CR completers compared to non-completers. This observation may have been due to a ceiling effect within the 5-meter gait speed and balance tests. A large group of individuals obtained the highest achievable score at both the baseline and 1-year post-operative time points; therefore, diminishing the ability of the SPPB to detect any potential benefit of CR programming.

Lastly, the data indicate that only the overall physical activity domain of the five FFI domains demonstrated a significant change in CR completers compared to non-completers. The increased sensitivity to change within the physical domain of frailty is supported in the literature, which suggests that gait velocity and timed-up-and-go test can be sensitive to change over a 2-week multidisciplinary intervention among frail elderly adults [44]. This demonstrates that either the physical domain of frailty was the only FFI domain that is sensitive enough to change over time or that the physical domain was the only modifiable domain assessed. Future research will need to determine which option is the case. 

### 4.3. CR Attendance

The Canadian Cardiovascular Society CR quality indicator regarding program adherence recommends that CR participants attend all sessions throughout the program to maximize favorable outcomes [22]. Given the dose–response relationship with CR attendance and improved outcomes such as mortality, it is possible that CR programming may be able to modify frailty status, particularly among CR attenders with high adherence [45]. Although a correlation between baseline CFS and CR attendance did reach statistical significance (*p* = 0.02), similar correlations between attendance and the other three frailty scores were not identified. Therefore, future research will need to clarify the impact that program adherence has for influencing frailty status. 

### 4.4. Limitations

This study is retrospective in nature, which limited our ability to select the collected variables at the baseline and 1-year post-operative time points. For example, we did not have access to 6-minute walking test data, which provides a strong indication of the response to medical interventions amongst patients with moderate–severe heart and lung disease [46]. The 6MWT may have some value for assessing changes in functional status for people with varying levels of frailty and aerobic fitness [47] and for patients who participate in prehabilitation programming before cardiac surgery [48].There is potential sample bias as only people who had a baseline frailty assessment and an in-clinic 1-year post-operative frailty assessment in the original prospective cohort study were included in this analysis. However, the only baseline characteristic difference between the included and excluded participants was comorbid diabetes. Frail individuals may have chosen not to participate in CR entirely. As such, these data may have underestimated the problem of CR not serving frail populations adequately. Additionally, 82% of our cohort were elective cardiac surgery patients, with the remaining 18% urgent cardiac patients. Therefore, the results of this study are limited regarding generalizability to those requiring urgent or emergent cardiac surgical procedures. On average, a median 261 (226–321) days elapsed between the final CR session attended and our 1-year post-operative frailty assessment. This period of time is significant considering that there is some literature supporting the fact that physical activity, among other variables, may dissipate over time following CR program conclusion [4,49,50]. More specifically, physical activity appears to peak one month into CR programming and returns to baseline levels at the 6-month and 12-month post CR [51]. 

## 5. Conclusions

This study demonstrated that cardiac surgery patients who were deemed to be frail pre-operatively were significantly less likely to complete CR post-operatively compared to their more robust counterparts. However, neither CR completion nor CR attendance was associated with a significant change in frailty over time. The MFC frailty domains of cognitive impairment and low physical activity in addition to the FFI physical domain of frailty were significantly improved among CR completers when compared to CR non-completers. This finding suggests that some domains of frailty may be more sensitive to change over time than others. The component analysis of frailty assessment tools indicates that there is an urgent need to refine existing frailty tools so they have the sensitivity to assess changes in frailty over time. 

## Figures and Tables

**Figure 1 jcm-07-00560-f001:**
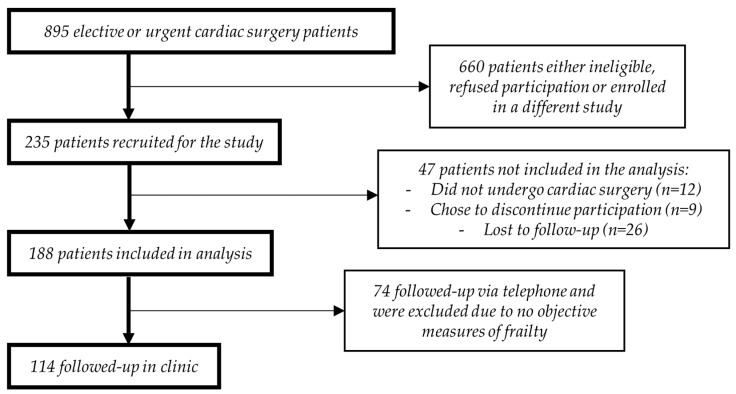
Participant flow diagram.

**Figure 2 jcm-07-00560-f002:**
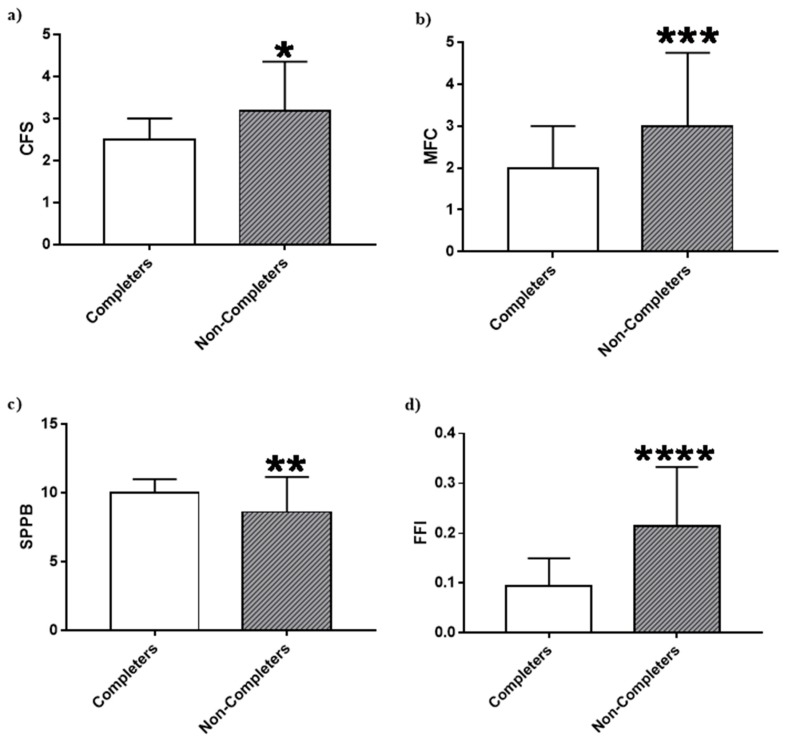
(**a**) Pre-operative CFS scores among CR completers and non-completers; (**b**) Pre-operative MFC scores among CR completers and non-completers; (**c**) Pre-operative SPPB scores among CR completers and non-completers; (**d**) Pre-operative FFI scores among CR completers and non-completers. Values are median ± interquartile range. Completers *n* = 48; non-completers *n* = 66. Statistical comparisons were calculated using a non-parametric Mann-Whitney test. * Non-completers different from completers (*p* < 0.05), ** Non-completers different from completers (*p* < 0.01), *** Non-completers different from completers (*p* < 0.001), **** Non-completers different from completers (*p* < 0.0001). A lower score for the CFS, MFC, and FFI signifies an individual who is less frail. However, the opposite is true for the SPPB, where a higher score signifies an individual who is less frail. CFS, Clinical Frailty Scale; MFC, Modified Fried Criteria; SPPB, Short Physical Performance Battery; FFI, Functional Frailty Index.

**Figure 3 jcm-07-00560-f003:**
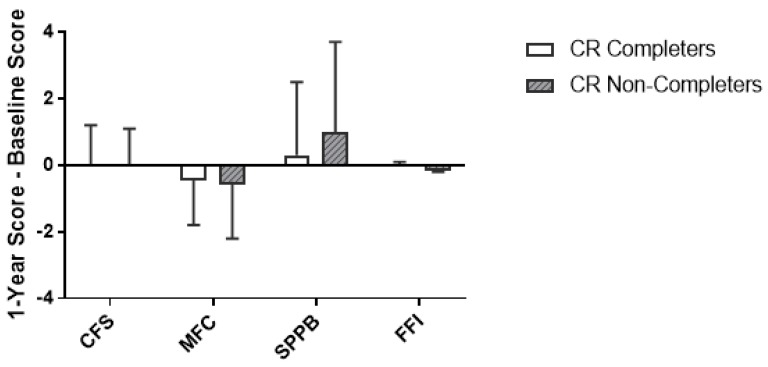
∆Frailty scores from baseline to 1-year post-operatively. Values are mean ± standard deviation. Completers *n* = 48; non-completers *n* = 66. Statistical comparisons were calculated using a non-parametric Mann-Whitney test. CFS, Clinical Frailty Scale; MFC, Modified Fried Criteria; SPPB, Short Physical Performance Battery; FFI, Functional Frailty Index.

**Table 1 jcm-07-00560-t001:** Frailty measurement tools and cut points.

Tool	Variables Considered	Frailty Cut Point
CFS	Subjective 9-point scale	≥4 points out of 9
MFC	Slowness, weakness, weight loss, exhaustion, depression, low physical activity, cognitive impairment	≥3 of the 7 variables present
SPPB	5 m gait speed, balance tests, repeated chair stand test	≤9 points out of 12
FFI	25 separate variables (Table A1)	deficits/variables ≥0.25

CFS, Clinical Frailty Scale; MFC, Modified Fried Criteria; SPPB, Short Physical Performance Battery; FFI, Functional Frailty Index.

**Table 2 jcm-07-00560-t002:** Characteristics comparing CR completers to non-completers.

	CR Completers (*n* = 48)	CR Non-Completers (*n* = 66)	*p*-Value
**Demographics**			
Age	70.5 (66–72)	71.5 (66.3–78)	0.08
Sex (Female)	18 (38%)	24 (36%)	0.29
BMI (kg/m^2^)	29.0 (25.0–31.6)	28.3 (25.4–32.2)	0.90
Lives Alone	6 (13%)	20 (30%)	0.02
Education (College or more)	25 (52%)	23 (35%)	0.07
Smoker (Never smoked)	19 (40%)	28 (42%)	0.71
**Pre-Surgery Risk**			
EuroSCORE II	1.26 (1–2.1)	1.77 (1.2–3.0)	0.07
**Comorbidities**			
Previous MI	11 (23%)	23 (35%)	0.17
CHF	23 (48%)	33 (50%)	0.70
Diabetes	6 (13%)	23 (35%)	0.006
CRF	1 (2%)	3 (5%)	0.48
COPD	2 (4%)	11 (17%)	0.04
Depression	5 (10%)	8 (12%)	0.78
**Surgical Parameters**			
Surgery Type			0.19
Isolated CABG	23 (48%)	29 (44%)	
Isolated Valve	11 (23%)	18 (27%)	
CABG + Valve	8 (17%)	17 (26%)	
Other	6 (12%)	2 (3%)	
ICU Length of Stay (days)	1 (1–2.25)	1 (1–3)	0.39
Length of Hospital Stay (days)	6 (5–8.5)	10 (6–14)	0.002

Continuous variables expressed as median (interquartile range) and categorical variables expressed as n (%). The Mann-Whitney test compared continuous variables, Chi-Square Test compared categorical variables. BMI, Body Mass Index; EuroSCORE II, European System for Cardiac Operative Risk Evaluation; MI, Myocardial Infarction; CHF, Chronic Heart Failure; CRF, Chronic Renal Failure; COPD, Chronic Obstructive Pulmonary Disorder; CABG, coronary artery bypass graft; ICU, intensive care unit.

**Table 3 jcm-07-00560-t003:** Change in frailty domains between baseline and 1-year post-operative.

	CR Completers (*n* = 48)		CR Non-Completers (*n* = 66)		*p*-Value
	**Baseline**	**1-Year**	**Baseline**	**1-Year**	
**MFC**					
Slowness (5-meter gait speed, s)	4.6 (3.8–5.6)	4.5 (4–5.4)	5 (4.2–6.3)	5.1 (4.7–5.7)	0.46
Weakness (grip strength; kg)	36.5 (25.5–41.3)	32 (22.5–41)	30 (20–40)	27 (18–37.5)	0.72
Weight loss in the past year (kg)	1.3 (0–4.5)	0 (0)	4.5 (2–9.3)	0 (0)	0.50
Exhaustion (CESD)	0 (0–2)	0 (0–1)	2 (0–3)	2 (0–3)	0.47
Depression (HADS)	2 (1–4)	1 (1–2)	3 (1–6)	3 (1–5)	0.32
Cognitive impairment (MOCA)	25 (23–27)	25 (22–28)	24 (21–27)	23 (18.3–25.8)	0.005
Low physical activity (Paffenbarger, kcal/wk)	437.5 (155–886)	1591 (672–3150)	96 (28.8–338.8)	658 (215.8–2105.8)	0.04
**SPPB**					
5-meter gait speed (points)	4 (4)	4 (4)	4 (4)	4 (4)	0.69
Balance (points)	4 (4)	4 (4)	4 (2.3–4)	4 (3–4)	0.06
Repeated chair stand (points)	2 (1–3)	3 (1.8–4)	2 (1–3)	3 (1.3–4)	0.87
**FFI**					
Physical	0.2 (0.1–0.3)	0.05 (0–0.17)	0.35 (0.2–0.45)	0.1 (0.05–0.31)	0.009
Functional	0 (0)	0 (0)	0 (0–0.05)	0 (0)	0.28
Nutrition and exhaustion	0.1 (0–0.2)	0 (0–0.2)	0.2 (0.1–0.4)	0.2 (0.03–0.2)	0.18
Quality of life	0.2 (0.2–0.3)	0.2 (0.1–0.3)	0.35 (0.2–0.78)	0.2 (0.2–0.3)	0.18
Mood and cognition	0.33 (0–0.33)	0.33 (0–0.33)	0.33 (0–0.33)	0.33 (0.33–0.33)	0.62

Continuous variables expressed as median (interquartile range). *p*-values indicate the difference in change between baseline and 1-year for CR completers and non-completers. The FFI domains include the following variables: Physical: balance, gait speed, chair stand, timed up-and-go, physical activity; Functional: help eating, dressing, cleaning, bathing, toileting, shopping, cooking, driving, medicating, banking; Nutrition and Exhaustion: 2-item CESD, past 3 month food decline, weight loss in the past 3 and 12 months; Quality of life: rating of own health, falls efficacy scale; Mood and cognition: depression, anxiety, MOCA. MFC, Modified Fried Criteria; SPPB, Short Physical Performance Battery; FFI, Functional Frailty Index; CESD, Center for Epidemiologic Studies Depression Scale; HADS, Hospital Anxiety and Depression Scale; MOCA, Montreal Cognitive Assessment.

**Table 4 jcm-07-00560-t004:** Correlations between CR attendance and frailty.

	CFS			MFC			SPPB			FFI		
	Baseline	1-Year	Delta	Baseline	1-Year	Delta	Baseline	1-Year	Delta	Baseline	1-Year	Delta
*r_s_*	−0.29	−0.24	0.062	−0.15	−0.082	0.072	0.025	0.16	0.15	−0.23	−0.21	0.0049
*p*-value	0.02	0.06	0.64	0.25	0.53	0.58	0.85	0.23	0.26	0.07	0.11	0.97

Spearman correlations are shown. CFS, Clinical Frailty Scale; MFC, Modified Fried Criteria; SPPB, Short Physical Performance Battery; FFI, Functional Frailty Index.

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
