# Peer review of "Pre-Operative Frailty Status Is Associated with Cardiac Rehabilitation Completion: A Retrospective Cohort Study"

_jcm, 2018, doi:10.3390/jcm7120560_

Reviewer 1 Report

Kimber and colleagues examined the frailty score before and after undergoing cardiac surgery. They found that 42% (48/114) completed cardiac rehabilitation (this is the primary end-point), and the pre-operative frailty score determined the post-operative status. Backgrounds of completers were characterized by living with family, without having diabetes mellitus and COPD. They discussed the assessment tool may have led to affect the results. It is interesting, but there are several concerns to be addressed to strengthen this manuscript.

1) It is unclear how to define the completers/ non-completers of cardiac rehabilitation. In addition, assess why non-completers did not accomplish the cardiac rehabilitation.

2) It is uncertain the content of rehabilitation program they underwent. Did they complete the rehabilitation throughout 1 year?

3) It is unclear that cardiac rehabilitation also affected the mortality and the readmission due to cardiovascular event, incidence of pneumonia, and so on in this cohort.

4) Explain the meaning, “CR swipe” more minutely somewhere in this manuscript.

Author Response

Point 1: It is unclear how to define the completers/ non-completers of cardiac rehabilitation. In addition, assess why non-completers did not accomplish the cardiac rehabilitation.

Response 1: The definition used for CR completion in this study, based on the Canadian Cardiovascular Society's definition, is included in section 2.3.2 when our primary outcome is discussed. Unfortunately, we did not have access to information on why participants did not complete beyond the definition of non-completion identified in the paper as this is a provincially-run cardiac rehab program and this information is not captured.

Point 2: It is uncertain the content of rehabilitation program they underwent. Did they complete the rehabilitation throughout 1 year?

Response 2: Further elaboration on the content of the cardiac rehab program has been added to the manuscript in the methods section (2.3). Briefly, the provincially-run cardiac rehab programs in Manitoba follow the guidelines outlined by the Canadian Associaton of Cardiovascular Prevention and Rehabilitation (cited in paper).

Point 3:  It is unclear that cardiac rehabilitation also affected the mortality and the readmission due to cardiovascular event, incidence of pneumonia, and so on in this cohort.

Response 3: Unfortunately, our sample size was too small to perform analyses on outcomes such as those you have listed above. These would be important variables for future research to examine.

Point 4: Explain the meaning, “CR swipe” more minutely somewhere in this manuscript.

Response 4: CR swipe has been more minutely defined in section 2.4.2. Following this definition, this variable will just be referred to as "attendance" in the manuscript. 

Reviewer 2 Report

The purpose of this retrospective cohort study was to determine if pre-operative frailty scores impacted post-operatively CR completion and if CR completion influenced frailty scores in 114 cardiac surgery patients. Frailty was assessed with the use of the Clinical Frailty Scale (CFS), the Modified Fried Criteria (MFC), the Short Physical Performance Battery (SPPB), and the Functional Frailty Index (FFI). CR non-completers were more frail than CR completers at pre-operative baseline based on the CFS (p=0.01), MFC (p<0.001), SPPB (p=0.007), and the FFI (p<0.001). A change in frailty scores from baseline to 1-year post-operation was not detected in either group using any of the 4 frailty assessments. However, greater improvements from baseline to 1-year post-operation in two MFC domains (cognitive impairment and low physical activity) and the physical domain of the FFI were found in CR completers as compared to CR non-completers. The Authors suggested that pre-operative frailty assessments could identify participants who are less likely to attend and complete CR. The data also suggest that physical domains of frailty function appear to be more sensitive to change following CR than other domains.

The manuscript is of interest and the topic is relevant for cardiac rehabilitation.

Several points should be described.

First of all I do not understand when started cardiac rehabilitation program. The time interval between the cardiac surgery intervention and rehabilitation should be described. Moreover, no mention of cardiac rehabilitation program was described. Is an hospital based program?

The Rehabilitation program in several nation is mandatory soon after cardiac surgery (1 week from the discharge) in order to capture the frailest part of patients with the aim to reduce complication and readmission to the hospital for this vulnerable elderly population.

What do you mean for stress test (6minute- walking test, ergometry stress test, ergospirometry ???).

If you selected your measure of outcome on the basis of the capacity to perform an ergometric stress test you probably loose the frailties part of your patients.   

You stated pre-operatively evaluation, how do you performed frailty evaluation in urgent

Procedures?.

I suggest to present also mean data for CR swipe card access in order to better understand the data.

The selection of the sample should be better defined. It is difficult to accept only 114 patients from an original sample of 895. This could hide a selection bias and criteria selection should be better analyzed. No mention of death. The study identify a sample of not so frail population considering a mean value of SPPB of 10, this could reduce the positive effect of cardiac rehabilitation because a ceiling effect related to variable.

Author Response

Point 1:  First of all I do not understand when started cardiac rehabilitation program. The time interval between the cardiac surgery intervention and rehabilitation should be described. Moreover, no mention of cardiac rehabilitation program was described. Is an hospital based program?

Response 1: Time interval between hospital discharge and rehabilitation initiation has been added to section 3.1. Description of the rehab program itself has been added to the methods section (2.3).

Point 2:  What do you mean for stress test (6minute- walking test, ergometry stress test, ergospirometry ???).

Response 2: Standard stress test for cardiac rehab in Manitoba is a modified Bruce protocol. This detail has been added in section 2.4.2.

Point 3:  If you selected your measure of outcome on the basis of the capacity to perform an ergometric stress test you probably loose the frailties part of your patients.   

Response 3:  Ability to perform a stress test is a requirement of cardiac rehabilitation participation in Manitoba. All cardiac surgery patients are automatically referred to cardiac rehab, but it is their choice whether or not to attend (now described in section (2.3). As such, it is possible that some frail individuals may have chosen not to participate. This note has been added to our limitations section. If anything, the inclusion of these individuals would only strengthen the message that cardiac rehab is not serving frail populations adequately.

Point 4: You stated pre-operatively evaluation, how do you perform frailty evaluation in urgent procedures.

Response 4:  18% of our overall population was composed of urgent cardiac patients as opposed to elective procedures (82%). These participants were still able to perform the testing necessary to capture frailty measures prior to their surgical procedures.

Point 5: I suggest to present also mean data for CR swipe card access in order to better understand the data.

Response 5:  This information has been added to section 3.1.

Point 6:  The selection of the sample should be better defined. It is difficult to accept only 114 patients from an original sample of 895. This could hide a selection bias and criteria selection should be better analyzed. No mention of death. The study identify a sample of not so frail population considering a mean value of SPPB of 10, this could reduce the positive effect of cardiac rehabilitation because a ceiling effect related to variable.

Response 6:  Further detail regarding study flow has been added to section 3.1. The choice of some frail individuals to not participate in the study may have resulted in our study actually underestimating the problem of cardiac rehab not serving frail populations adequately. This has been added to our limitations section. 

Outcome variables such as mortality or rehospitalization could not be analyzed due to our sample size. 

It is possible that examination of change in frailty was affected by a ceiling effect. This note was made on line 49-50 in section 4.1. 

Round  2

Reviewer 2 Report

The manuscript is improved and all point of criticism were discussed.

I have only a last finding. The “6 Minute walking test” is an important sub-maximal exercise used in CR to assess the functional capacity, especially in elderly – frail patient. I think is important to discuss this point and to include this reference. Cacciatore F et al. Six-minute walking test but not ejection fraction predicts mortality in elderly patients undergoing cardiac rehabilitation following coronary artery bypass grafting. Eur J Prev Cardiol. 2012 Dec;19(6):1401-9.

Author Response

Point 1: I have only a last finding. The “6 Minute walking test” is an important sub-maximal exercise used in CR to assess the functional capacity, especially in elderly – frail patient. I think is important to discuss this point and to include this reference. Cacciatore F et al. Six-minute walking test but not ejection fraction predicts mortality in elderly patients undergoing cardiac rehabilitation following coronary artery bypass grafting. Eur J Prev Cardiol. 2012 Dec;19(6):1401-9.

Response 1We agree that the 6MWT is an important CR indicator. We have now added text to the manuscript to identify that we did not have access to 6MWT data. We have cited literature describing this as well as the possible utility that the 6MWT may have for assessing changes in frailty status. We also reviewed the citation provided by the reviewer (Cacciatore et al.) and decided not to cite in the current manuscript; rather, we have cited the ATS Committee on Proficiency Standards for Clinical Pulmonary Function Laboratories. ATS statement: Guidelines for the six-minute walk test, as that clinical guideline is directly relevant to the information described.